# HFENet: Hierarchical Feature Extraction Network for Accurate Landcover Classification

**Di Wang [1], Ronghao Yang [1,2,*] , Hanhu Liu [1], Haiqing He [3] , Junxiang Tan [1], Shaoda Li [1], Yichun Qiao [1], Kangqi Tang [1] and Xiao Wang [1]**

1   Department of Surveying & Mapping Engineering, College of Earth Sciences,
    Chengdu University of Technology, Chengdu 610059, China
2   Chengdu Technical Center of Xinhui Zhiyun Group Co., Ltd., Chengdu 610058, China
3   School of Geomatics, East China University of Technology, Nanchang 330013, China
*   Correspondence: yangronghao@cdut.edu.cn

**Abstract:** Landcover classification is an important application in remote sensing, but it is always a challenge to distinguish different features with similar characteristics or large-scale differences. Some deep learning networks, such as UperNet, PSPNet, and DANet, use pyramid pooling and attention mechanisms to improve their abilities in multi-scale features extraction. However, due to the neglect of low-level features contained in the underlying network and the information differences between feature maps, it is difficult to identify small-scale objects. Thus, we propose a novel image segmentation network, named HFENet, for mining multi-level semantic information. Like the UperNet, HFENet adopts a top-down horizontal connection architecture while includes two improved modules, the HFE and the MFF. According to the characteristics of different levels of semantic information, HFE module reconstructs the feature extraction part by introducing an attention mechanism and pyramid pooling module to fully mine semantic information. With the help of a channel attention mechanism, MFF module up-samples and re-weights the feature maps to fuse them and enhance the expression ability of multi-scale features. Ablation studies and comparative experiments between HFENet and seven state-of-the-art models (U-Net, DeepLabv3+, PSPNet, FCN, UperNet, DANet and SegNet) are conducted with a self-labeled GF-2 remote sensing image dataset (MZData) and two open datasets landcover.ai and WHU building dataset. The results show that HFENet on three datasets with six evaluation metrics (mIoU, FWIoU, PA, mP, mRecall and mF1) are better than the other models and the mIoU is improved 7.41–10.60% on MZData, 1.17–11.57% on WHU building dataset and 0.93–4.31% on landcover.ai. HFENet can perform better in the task of refining the semantic segmentation of remote sensing images.

**Keywords:** landcover classification; semantic segmentation; hierarchical feature extraction (HFE); multi-level feature fusion (MFF)

## 1. Introduction

Thanks to the rapid development of aerospace technology, communication technology and information processing technology, people have entered the era of remote sensing big data. How to fully explore and mine the growing remote sensing image information has become an urgent problem to be solved [1–3]. Landcover classification in remote sensing is a basic and important task in remote sensing big data processing [4–6], and it is also a basic work for ecological environment protection [7,8], urban planning [9], geological disaster monitoring [10,11], and other fields.

At present, landcover classification in remote sensing images mainly adopts machine learning methods, including shallow machine learning and deep learning. Among them, shallow machine learning methods are mainly represented by Random Forest (RF) [12], Support Vector Machine (SVM) [13], etc., which are based on manually-extracted objects

such as color, texture, geometric shape and spatial structure, and other feature information. Landcover classification is achieved by learning classification rules from supervised information [14]. Deep learning methods automatically extract low-level image features of objects from images by building deep networks, and combining them into high-level abstract features, whereby higher classification accuracy can be achieved, which have become the mainstream methods for remote sensing image landcover classification research [15,16].

Since remote sensing images not only have the phenomena of "inter-class similarity and intra-class variance", but also have large scale differences between objects of the same class, which make the automatic classification of remote sensing images have the problems of confusing the classification of similar features and difficulty in identifying small-scale features [16]. To address the confusing problem of similar features classification, scholars [17–20] used Pyramid Pooling Module (PPM), Attention Mechanism (AM), and other methods to model the spatial location and channel relationship. These modules extract the contextual information of features and mutual information between channels, improve the model's ability to model and understand complex scenes, and reduce confusing information interference. However, these methods are aimed at the feature extraction of the lowest level features, and do not make full use of the spatial position information contained in the top-level features. They are more suitable for the recognition of large-scale objects, while the recognition effect of small-scale objects is not ideal.

To solve the problem that small-scale features are difficult to be recognized, the current methods [21–24] mostly use the combination of feature extraction and feature fusion to recover the detailed information of images step by step, among which the representative networks are U-Net [25], FPN [26], UperNet [27], Swin [28] and Twins [29], etc. In the research of remote sensing image landcover classification, these methods are often used to improve the recognition ability of multi-scale features. In order to improve the effect of feature extraction, Dalal AL-Alimi [30] proposed a method combining pyramid extraction network and SE attention mechanism, which can reduce the loss of small objects by selectively retaining the useful information in the feature map through SE. However, the feature map is not fully mined, resulting in the deviation of the detected anchor frame. Wenzhi Zhao [31] uses graph convolution to extract the bottom features of the network, which is used to capture the long-term dependence in the network and improves the ability to obtain the network context information. However, it ignores the spatial position information contained in the top feature information, resulting in an inaccurate outline of the recognized features. Jianda Cheng [32] uses a capsule network instead of ResNet for feature extraction, which can enhance the network's hierarchical understanding of the whole and part of the object and is more conducive to the network's modeling of the object. However, the advantages of this global and local representation can be more obvious in the recognition of large-scale objects and have little impact on the recognition of small-scale objects. In view of the improvement of the effect of feature fusion method, Yong Liao [33] uses the attention mechanism and residual connection to fuse multi-scale features, which can improve the ability of the network to extract low-level feature information and high-level semantic information. However, when using the attention mechanism to fuse features, it mainly operates on the underlying features, ignoring the difference between the top-level features and the underlying features, and increasing the risk of small-scale objects being ignored. Qinglie yuan [34] uses the residual branch network to assist the backbone network in feature transformation, which can enhance the multi-modal data fusion ability of the network. However, this method of realizing adjacent feature fusion through simple element addition ignores the difference of information between feature maps, resulting in the inability to accurately extract low-level semantic information such as the position of the object, and it is difficult to identify small-scale targets [22,35].

To solve the above-mentioned problems, based on UperNet [27], this paper proposes a semantic segmentation network, named HFENet, for mining multi-level semantic information, to solve the problems that similar features are easy to be confused and small-scale features are difficult to be identified in remote sensing image landcover classification, so as to improve the accuracy of remote sensing image landcover classification. The main contributions of this paper include:

(1) A Hierarchical Feature Extraction (HFE) strategy is proposed. According to the difference of the information contained in the top-level and bottom-level network feature maps, the strategy adopts specific information-mining methods in different network layers to extract the spatial location information, channel information, and global information contained in the feature maps, so as to improve the information mining ability of the network.

(2) A Multi-level Feature Fusion (MFF) method is proposed. Aiming at the fusion problem of multiple feature maps with size and semantic differences, this method adopts the method of up sampling the input feature maps step by step and re-weighting them according to the channels, so as to reduce the impact caused by the difference of semantic information, improve the attention of the network to the spatial location information, and enhance the feature expression ability of the network.

(3) A Hierarchical Feature Extraction Network (HFENet) model is proposed, which includes HFE and MFF modules. First, the HFE strategy is used to fully mine the information of feature maps, and then the MFF method is used to enhance the expression of feature information, so as to improve the recognition ability of the network to the easily confused and small-scale features and achieve the result of accurate surface coverage classification.

(4) The effectiveness of the two modules proposed in our framework is verified by ablation experiments; the effectiveness of our proposed HFENet was demonstrated by performing landcover classification/image segmentation on three remote sensing image datasets and comparing it with the state-of-the-art models (PSPNet [17], DeepLabv3+ [36], DANet [18], etc.).

The rest of this paper is organized as follows. Section 2 introduces related research work, mainly reviews the development of semantic segmentation in the field of remote sensing image landcover classification in recent years and focuses on the methods based on deep learning. Section 3 elaborates on the structure of the proposed HFENet and details the design ideas of the proposed HFE and MFF modules. Section 4 gives the experimental details and results on a self-labeled dataset (MZData) and two public datasets (landCover.ai, WHU Building dataset) [37,38]. In Section 5, a comprehensive analysis is performed for the obtained results. Section 6 contains a discussion.

## 2. Related Work

In this section, we first introduce the research of deep Semantic Segmentation Network (SSN) in remote sensing image landcover classification, and then discuss the research of AM in image SSN.

### 2.1. Research on Landcover Classification with Semantic Segmentation Network

SSN is a hot research method in remote sensing image landcover classification. By automatically extracting object features from original images, it is more beneficial to mine high-level semantic information and achieve high-precision pixel-level classification [33,34]. Classical SSNs include FCN [39], U-Net [25], DeepLab series of networks [36,40–42] and PSPNet [17], which are aimed at medicine, autonomous driving and other fields, respectively. For problems such as receptive field, multi-scale features, edge recognition refinement and global context information, corresponding solutions are proposed to improve the feature extraction and feature expression capabilities of the network.

To address the problem of difficult recognition of small-scale surface objects in remote sensing images, Zheng et al. [43] proposed an end-to-end Edge-aware neural Network (EaNet) that captures rich multi-scale contextual information with strong continuous feature relationships by combining a Large Kernel Pyramid Pooling (LKPP) module; Wang et al. [44] proposed a bilateral perceptual network containing dependent paths and texture paths to fully capture long-term relationships and detailed information in VHR images; Cheng et al. [45] proposed a cascade segmentation refinement model (CascadePSP), which achieves refinement segmentation by aggregating the features extracted from different layers of the backbone network. To be able to refine the object boundary information, Zhou et al. [46] incorporated the edge detection task and the semantic segmentation task into the same framework. Guo et al. [47] proposed an end-to-end double-gate fusion network (DGFNet), which effectively extracts both low-level spatial information and high-level semantic information of the image. These studies all combine semantic information from different layers in the network to obtain finer pixel-level classification results; however, since they often use a single approach to extract semantic features at different layers, they do not consider the variability of semantic features at different layers, which may lead to the lack of extraction results.

### 2.2. Attention Mechanisms in Image Semantic Segmentation Network

AM is a process of feature selection [48]. It promotes the feature extraction and expression ability of the entire network by shifting attention to the most relevant features and ignoring irrelevant parts, so as to efficiently analyze and understand complex scenes [49–51]. In computer vision, the AM extracts information that is more beneficial to the task through adaptive weighting according to the input image information and has achieved good results in many visual tasks [28,52,53]. Four basic types of attention can be classified according to the dimensions in which they act [50]: Channel Attention, Spatial Attention, Temporal Attention, and Branching Attention. Since channel and spatial location information are the basic attributes of images, the channel and Spatial Attention Mechanism can encode and infer image features from two different dimensions, thereby enhancing the network's ability to learn feature information. Therefore, in image semantic segmentation, we usually focus on applications of Channel Attention and SA [52,54,55].

In terms of Channel Attention, Hu et al. [56] proposed a "Squeeze and Excite" module (SE), which compresses the global features of each channel through Global Average Pooling (GAP), using full connection to obtain the relationship between channels and improve the representation ability. Based on the SE idea, Woo et al. [52] used maximum pooling and average pooling instead of GAP to improve the representation of the network using shared MLPs for channel relation inference. In terms of Spatial Attention, Hou et al. [57] proposed Coordinate Attention (CA), which uses a specific convolutional approach to encode each channel in two spatial directions and uses convolution to generate an attention vector that can accurately acquire the location of the target, effectively enhancing the learning and expression abilities. Existing research shows that the AM can analyze the features most relevant to the current task, which can be used for judgment and decision-making of high-level semantic information [18,58,59]. However, since the high-level semantic information is usually extracted based on the level-by-level down sampling of the backbone network, it may lead to the loss of small-scale objects in the image and the precise position information between objects, so that the small-scale objects cannot be fully recognized.

## 3. Methods

In order to solve the problems of confusing the classification of similar features and difficult recognition of small-scale features in landcover classification, we propose an improved SSN based on UperNet, namely HFENet (as shown in Figure 1). In this section, we first introduce the structure of HFENet, and then introduce each part of the framework in detail, as well as the design ideas of each part.

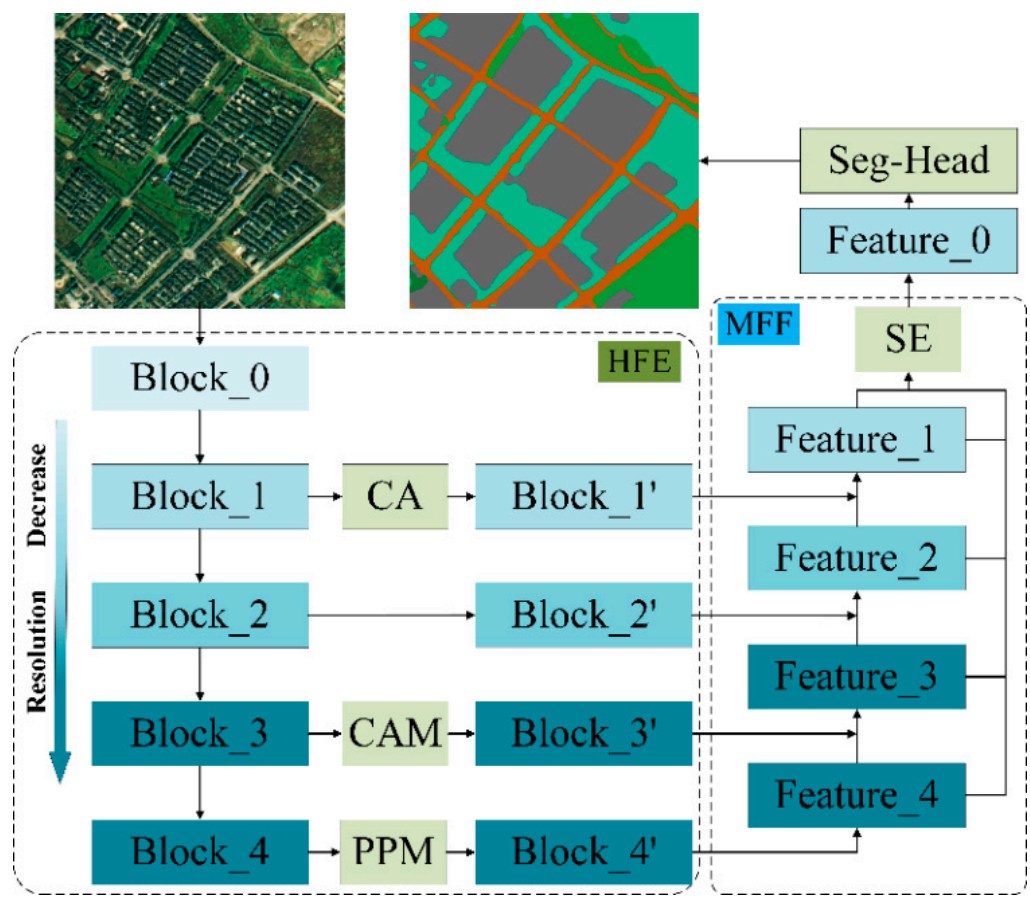

**Figure 1.** The overall framework of HFENet.

### 3.1. HFENet

The entire structure of HFENet is similar to that of UperNet. It is a top-down horizontal connection system (as shown in Figure 1), which consists of two parts: HFE and MFF. The HFE module takes the original image as input and obtains five different scale feature maps Block_$i$ ($i$ = 0, 1, 2, 3, 4) through the backbone network (generally using the ResNet series network), and then uses a specific operation (CA, CAM or PPM) for feature extraction, generating four feature maps Block_$i'$ ($i$ = 1, 2, 3, 4) containing different feature information and scales. The MFF module takes the output feature map of the HFE module as input. First, MFF transforms Block_4′ into Feature_4, upsamples Feature_4 to the same size as Block_3′, adds it to Block_3′ to generate Feature_3, and repeats this process to generate Feature_2 and Feature_1. Then, Feature_$i$ ($i$ = 2, 3, 4) is upsampled to the same size as Feature_1 and superimposed on it, and the SE is used for weight assignment to obtain the final feature map Feature_0. Finally, Feature_0 is input into the semantic segmentation classifier to obtain the final segmentation result. The detailed information about HFENet is shown in Algorithm 1.

---

**Algorithm 1:** Hierarchical Feature Extraction Network (HFENet).

---

**Input:** original image, backbone (ResNet)
**Output:** final segmentation result $P_r$
**Initialize:** random initialization of weights for **CA**, **CAM**, **PPM** and **SE**

1:     Extract feature maps **B** from original image through backbone: **B** = $[b_0, b_1, b_2, b_3, b_4]$
2:     Extract hierarchical feature map using specific operations from **B**: $b'_1 \leftarrow$ **CA**$(b_1)$, $b'_2 \leftarrow b_2$,
      $b'_3 \leftarrow$ **CAM**$(b_3)$, $b'_4 \leftarrow$ **PPM**$(b_4)$
3:     Unify the number of channels of the feature map by convolution operation (**Conv**):
      $f_4 \leftarrow$ **Conv**$(b'_4)$, $b'_3 \leftarrow$ **Conv**$(b'_3)$, $b'_2 \leftarrow$ **Conv**$(b'_2)$, $b'_1 \leftarrow$ **Conv**$(b'_1)$
4:     Up sample (**Up**) the feature maps and connect them with shortcuts step by step:
      $f_3 \leftarrow b'_3 +$ **Up**$(f_4)$, $f_2 \leftarrow b'_2 +$ **Up**$(f_3)$, $f_1 \leftarrow b'_1 +$ **Up**$(f_2)$
5:     Unify the shapes of the feature maps with up sample (**Up**): $f_4 \leftarrow$ **Up**$(f_4)$, $f_3 \leftarrow$ **Up**$(f_3)$,
      $f_2 \leftarrow$ **Up**$(f_2)$
6:     Concatenate (**Cat**) feature maps by channel and assign their weights with **SE**: $f_0 \leftarrow$ **SE**
      (**Cat**$(f_4, f_3, f_2, f_1)$)
7:     Obtain the final segmentation result $P_r$ through the semantic segmentation classifier
      (Seg_Head): $P_r \leftarrow$ **Seg_Head**$(f_0)$

---

### 3.2. Hierarchical Feature Extraction (HFE)

Since the underlying network features of a deep convolutional neural network contain rich spatial location information, and the top-level network features contain more high-level semantic information [47], using a single feature extraction method for different feature layers may lead to information loss and affect the accuracy of small objects recognition. In order to make better use of the feature information of different layers in the deep convolutional neural network, the strategies of using CA, CAM and PPM to pay attention to spatial location, channel relationship, and global information are designed respectively, so that the network can extract richer feature information and enhance the network's ability to recognize small objects.

In the HFE module (as shown in Figure 2), Block_1, Block_2, Block_3, and Block_4 respectively represent the feature maps output by different layers of the backbone network. In many state-of-the-art networks [17,59,60], they use Block_4 to extract multi-scale features information to improve the network's utilization of global context information; use Block_3 to classify, segment and calculate losses to assist network classification decisions and speed up network convergence speed. Block_1 is at the bottom of the network and is the main network layer representing low-level information of spatial location; Block_2 is between Block_1 and Block_3, mixing high-level semantic information and low-level spatial location information.

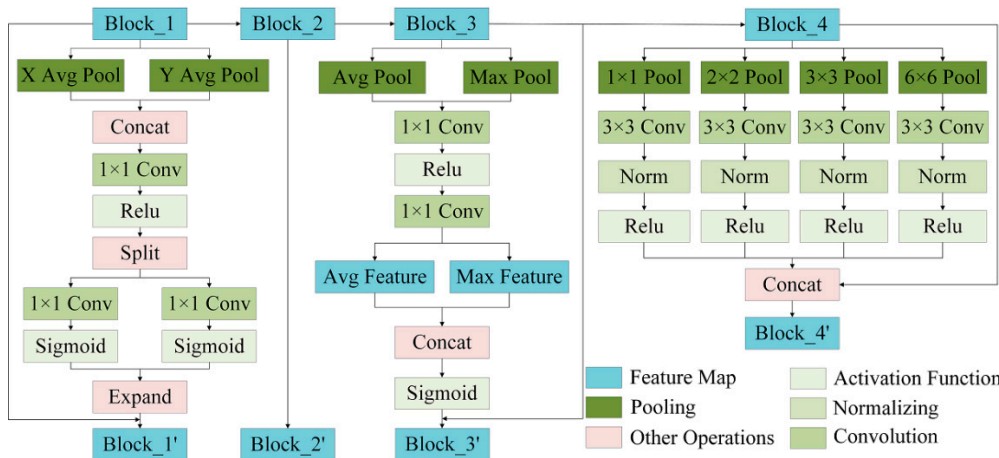

**Figure 2.** The detailed structure of Hierarchical Feature Extraction (HFE).

This paper adopts the PPM for Block_4. PPM obtains multi-scale feature information of feature maps by different pooling methods and aggregates the multi-scale information to obtain Block_4' with global, multi-scale information to enhance the network's ability to utilize global information. For Block_3, CAM [52] is introduced. First, CAM takes the feature map along the channel direction by maximum pooling and global pooling to obtain the salient information and background information on the feature map channels, and then uses MLP to model the two information to fully explore the interrelationship between each channel and improve the specific high-level semantic information expression, and finally sums with the input feature map to obtain Block_3' containing the channel relationships. CA [57] is introduced for Block_1. First, CA uses a specific convolution kernel to pool the different directions of the spatial dimension of the feature map and retains the most significant features of the image in the X and Y directions and aggregates them. Then, the 2D convolution is used to fuse the information of the feature maps in the X and Y directions to establish the spatial relationship of the feature maps. Finally, Block_1' containing positional relationship information is obtained by adding the input feature map, thereby improving the network's attention to specific spatial positional information. The specific process of HFE is illustrated in pseudocode Algorithm 2.

---

**Algorithm 2:** Hierarchical Feature Extraction (HFE).

---

**Input:** feature map $\mathbf{B} = [b_1, b_2, b_3, b_4]$
**Output:** hierarchical feature map $\mathbf{B'} = [b'_1, b'_2, b'_3, b'_4]$
**Initialize:** random initialization of weights for convolution operator (**Conv**)

1:  Conduct average pooling operation for $b_i$ along the X and Y axes respectively:
    $b_x \leftarrow$ **avg_pool_x** $(b_1)$, $b_y \leftarrow$ **avg_pool_y** $(b_1)$
2:  Concatenate (**Cat**) $b_x$ and $b_y$ by channel and combine them with $1 \times 1$ **Conv** operator:
    $b_{xy} \leftarrow$ **Conv** (**cat**$(b_x, b_y)$)
3:  Split $b_{xy}$ by channel: $(b'_x, b'_y) \leftarrow b_{xy}$
4:  Convolution operation is performed on $b'_x$ and $b'_y$ respectively to obtain respective position information: $b'_x \leftarrow$ **Conv** $(b'_x)$, $b'_y \leftarrow$ **Conv** $(b'_y)$
5:  Expand $b'_x$ and $b'_y$ respectively, and then multiply them to obtain a feature map with X and Y position information: $b'_1 \leftarrow b_1$ * **expand** $(b'_x)$ * **expand** $(b'_y)$

    // Computation for $b'_1$

6:  $b'_2 \leftarrow b_2$

    // Computation for $b'_2$

7:  Max and average pooling operations are performed on $b_3$ to obtain background information $b_a$ and saliency information $b_m$ respectively: $b_a \leftarrow$ **avg_pool** $(b_3)$, $b_m \leftarrow$ **max_pool** $(b_3)$
8:  Establish the relationships between the channels of $b_a$ and $b_m$ respectively by $1 \times 1$ **Conv** operator: $b'_a \leftarrow$ **Conv** $(b_a)$, $b'_m \leftarrow$ **Conv** $(b_m)$
9:  By concatenating $b'_a$ and $b'_m$ and multiplying with $b_3$, the channel relationship of the feature map is obtained: $b'_3 \leftarrow b_3$ * **cat** $(b'_a, b'_m)$

    // Computation for $b'_3$

10: Conduct average pooling operation on $b_4$ to obtain the global information $b_{1 \times 1}$ and the local information $b_{2 \times 2}, b_{3 \times 3}, b_{6 \times 6}$
11: Perform convolution operation on $b_{1 \times 1}, b_{2 \times 2}, b_{3 \times 3}$ and $b_{6 \times 6}$ respectively, and then concatenate them to obtain $b_m$ integrating global and local information:
    $b_m \leftarrow$ **cat**(**Conv**$(b_{1 \times 1})$, **Conv**$(b_{2 \times 2})$, **Conv**$(b_{3 \times 3})$, **Conv**$(b_{6 \times 6})$)
12: Multiply $b_4$ and $b_m$ to obtain a feature map containing multi-scale information: $b'_4 \leftarrow b_4$ * $b_m$

    // Computation for $b'_4$, end.

---

### 3.3. Multi-Scale Feature Fusion (MFF)

In order to fuse the high-level semantic information and spatial location information extracted by different network layers and improve the segmentation accuracy of remote sensing images in complex scenes, based on FPN [26], this paper designs a MFF module

(as shown in Figure 3). In this module, the output of the HFE module-Block_$i'$ ($i$ = 1, 2, 3, 4) is used as input, firstly, the number of feature channels of the four feature maps is unified, then the size of the four feature maps is unified and fused, and finally the fused feature maps are assigned weights to obtain the feature maps with specific attention information to enhance the network representation of low-level features.

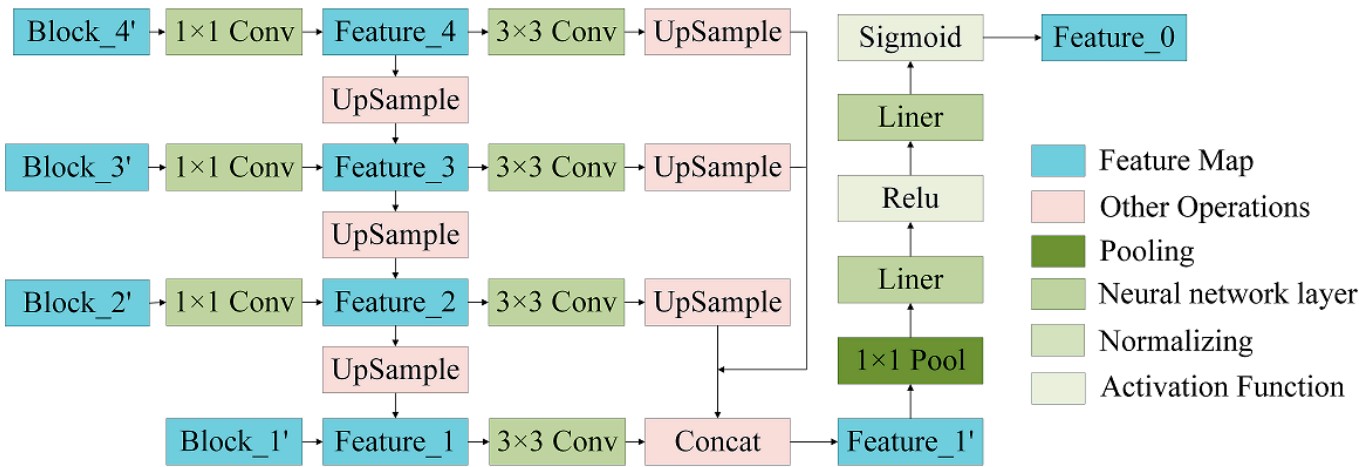

**Figure 3.** The detailed structure of the Multi-scale Feature Fusion (MFF).

In order to fuse the high-level semantic information and spatial location information extracted by different network layers and improve the segmentation accuracy of remote sensing images in complex scenes, based on FPN [26], this paper designs an MFF module (as shown in Figure 3). In this module, the output of the HFE module-Block_$i'$ ($i$ = 1,2,3,4) is used as input, firstly, the number of feature channels of the four feature maps is unified, then the size of the four feature maps is unified and fused, and finally the fused feature maps are assigned weights to obtain the feature maps with specific attention information to enhance the network representation of low-level features.

In MFF, the number of feature channels of the feature map is unified to be the same as Block_1' through operations such as 1*1 convolution, upsampling and residual connection, and four Feature_$i$ with the same number of feature channels are obtained ($i$ = 1, 2, 3, 4); and then unify the size of the feature map to the same size as Feature_1 through 3*3 convolution, up-sampling and residual connection and other operations, and connect the fused feature map Feature_1' in terms of channels. However, due to the difference of different levels of feature information extracted by the network, more weight is often given to high-level semantic information during classification. Therefore, in the classification task, the network will focus more on the feature expression of Feature_3 and Feature_4, and even ignore the low-level spatial location information of Feature_1 and Feature_2 (the experiments in this paper also prove that the information of Feature_1 is ignored), thereby reducing the segmentation performance. This paper introduces SE [56] for Feature_1'. When fusing multi-level features, it not only focuses on high-level semantic information, but also maintains a high degree of attention to the rich location information and texture information contained in the underlying network and redistributes the weight information of the feature map to obtain Feature_0. Algorithm 3 shows the specific operations and detailed process of computing MFF.

---

**Algorithm 3:** Multi-Scale Feature Fusion (MFF).

---

**Input:** feature maps: **B′** = [$b'_1, b'_2, b'_3, b'_4$]
**Output:** fused multi-scale feature map: $f_0$
**Initialize:** random initialization of weights for convolution operator (**Conv**), **F**= []

1:　　　**for** $i$ = 4 to 1 **do**
2:　　　　**if** $i$ == 4 **then**
3:　　　　　$b'_i \leftarrow$ **Conv**($b'_i$)　　　　　　　// Convolution operation on $b'_i$.
4:　　　　**else**
5:　　　　　$b'_i \leftarrow$ **Conv**($b'_i$) + **Up**($b'_{i+1}$)　　// Convolution operation on $b'_i$, up sample on $b'_{i+1}$.
6:　　　　**end if**
7:　　　**end for**

// Operations of unifying the number of channels of each feature map.

8:　　　**for** $j$ = 4 to 1 **do**
9:　　　　**if** $j$ == 1 **then**
10:　　　　　$f_j \leftarrow$ **Conv**($b'_j$)
11:　　　　**else**
12:　　　　　$f_j \leftarrow$ **Up**(**Conv**($b'_j$))
13:　　　　**end if**
14:　　　　**F**.append($f_j$)
15:　　　**end for**

// Operations of unifying the size of shapes of each feature map.

16:　　Fuse the feature maps $f_1, f_2, f_3$ and $f_4$ by **cat** operator: $f_f \leftarrow$ **cat**($f_1, f_2, f_3, f_4$)
17:　　Use **pool** operator to initialize the weights of channels of $f_f$, and then use **linear** operator to readjust the weights: $f_m \leftarrow$ **Linear**(pool ($f_f$))
18:　　Obtain the feature map $f_0$ fused by the redistributed weight: $f_0 \leftarrow f_f + f_f * f_m$

---

## 4. Experiments and Results

In this section, we focus on the effectiveness of HFENet. Firstly, we verify the role of HFE and MFF in the self-labeled dataset MZData, and then we verify the advantage of HFENet with the datasets of MZData, landcover.ai, and WHU building dataset. Next, we first introduce the datasets used and the parameters involved in the experiments, then explain the experimental design and analyze the experimental results in detail.

### 4.1. Experiments Settings

#### 4.1.1. Datasets

MZData

This dataset is a land-use/landcover classification dataset produced by a combination of manual interpretation and field survey data using fused satellite imagery of Gaofen-2 (GF-2). The spatial resolution of the image is 1 m, containing three bands of RGB. The original satellite images coverage area is Mianzhu City, Sichuan Province, China (Figure 4), located in the northwest of the Sichuan Basin, between 31°09′N–31°42′N and 103°54′E–104°20′E. The area is 1245.3 km$^2$. The city of Mianzhu contains both mountainous and plain terrain areas. Among them, the mountainous areas are mainly woodland, grassland and bare land; while the plain areas contain rich types of ground features, such as buildings, cropland and roads. In the plain area, there are abundant artificial features and large-scale differences between the same features, especially roads and buildings, which have high requirements on the recognition ability of small target objects. The land-use/landcover classification of this dataset contains eight categories, namely cultivated land, garden land, forest land, grassland, buildings, roads, water bodies, and bare land. In addition, in order to deal with the non-study area part introduced in the cropping process, a new category is added as background with all pixel values of 255, which will not have a significant impact on the algorithm classification results.

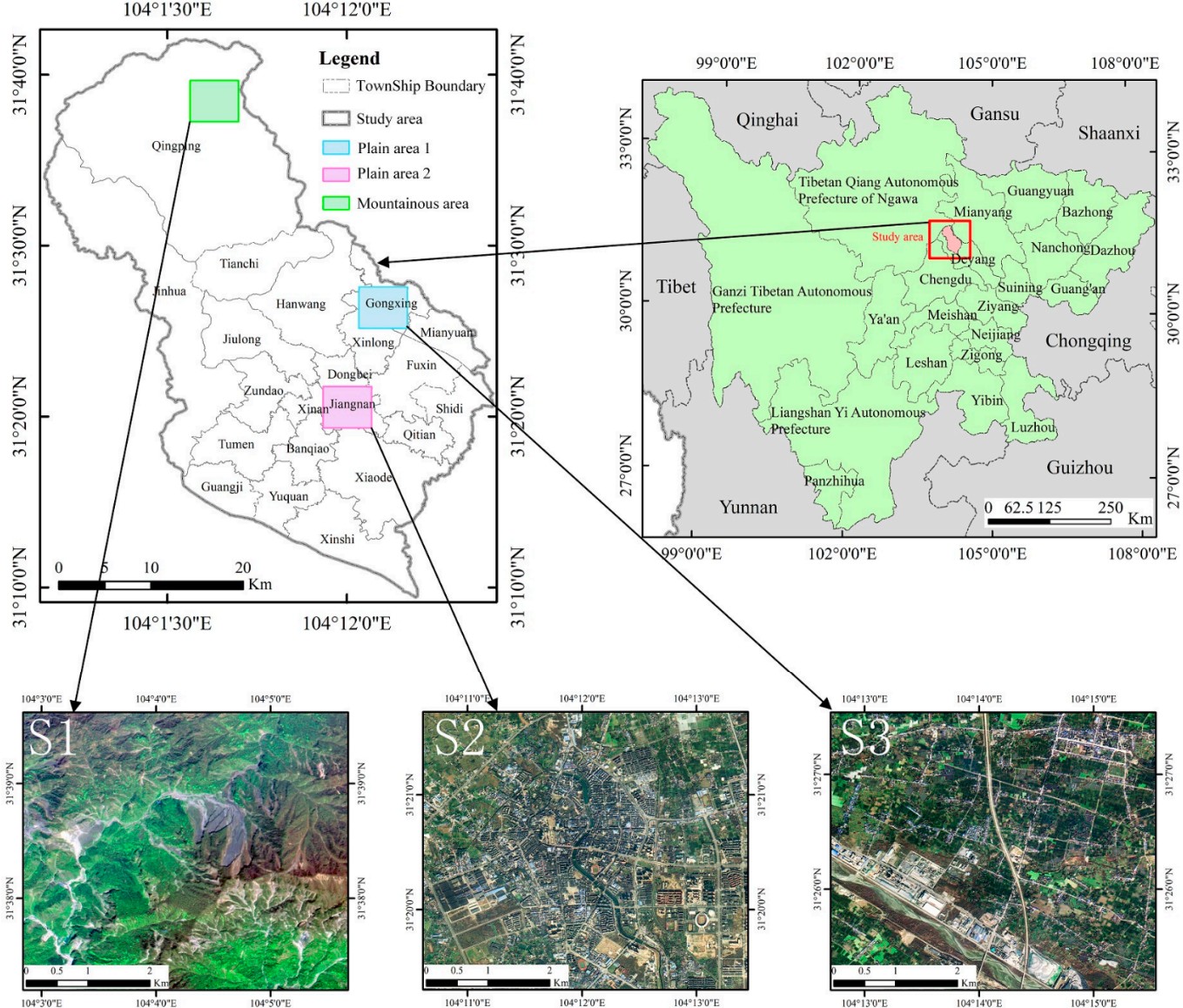

**Figure 4.** MZData geographical location and different typical terrain regions (**S1** represents mountainous area, **S2** and **S3** represent cities and villages in the plain area respectively).

According to the input requirements of the experimental network, the remote sensing images and interpretation results are sliced into sample images with a resolution of 512*512, and the sample pairs that are all background or contain clouds are manually removed to obtain 10,000 sample images; then the sample set is divided into training set, validation set and test set according to the ratio of 6:2:2 to establish a sample library of land-use/landcover classification data (as shown in Figure 5).

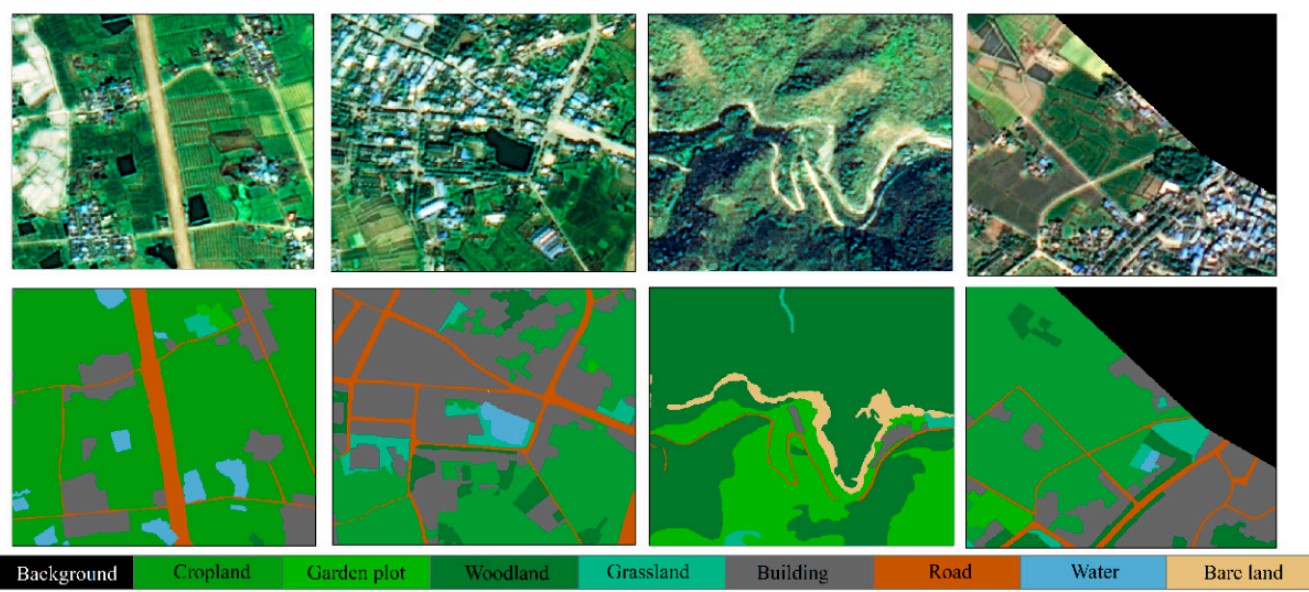

**Figure 5.** Some typical sample images of MZData and their corresponding labels.

LandCover.ai

The LandCover.ai dataset [37] is a dataset for aerial imagery landcover classification. The dataset covers an area of 216 km$^2$ and contains a total of 41 aerial images, which were taken in Poland and Central Europe. All image data only have three RGB bands. Among them, are 33 orthophotos with 25 cm realistic resolution for each pixel and eight orthophotos with 50 cm realistic resolution for each pixel. The dataset provides a detailed classification of landcover for the main areas of all images by means of manual interpretation, according to three feature categories of buildings, woodlands, and water, and one "other" category. Due to the high spatial resolution of the ground, the image characteristics of different objects are quite different and easy to distinguish.

According to the data set requirements, the data set is divided into 10,674 images with a resolution of 512 × 512, and the training set, validation set, and test set are divided according to the requirements, including 7470 training set images, 1602 validation set images and 1602 test set images, as shown in Figure 6.

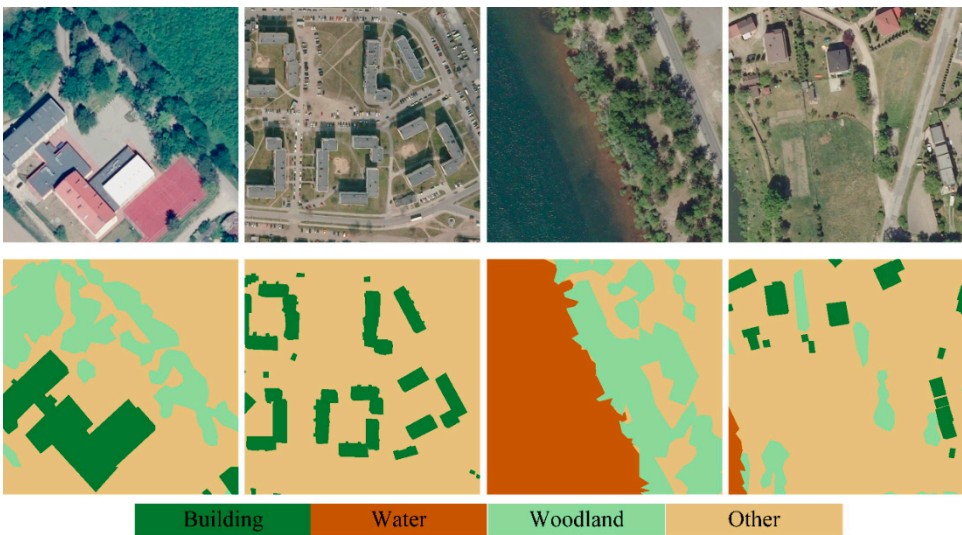

**Figure 6.** Some typical sample images of landcover.ai and their corresponding labels.

WHU Building Dataset

This dataset is an aerial image building dataset extracted by Wuhan University [38]. The image contains RGB three channels information, and the original ground resolution is 7.5 cm. By manually interpreting the building vector data of Christchurch, New Zealand, a data set covering an area of about 450 km$^2$ and 22,000 independent buildings was obtained. Due to the wide coverage area and the large number of buildings, the size and type of buildings vary greatly.

The data set contains 8188 images with 512 × 512 resolution and is divided into training set, validation set and test set. Among them, the training set contains 130,500 independent buildings in 4736 images, the validation set contains 14,500 independent buildings in 1036 images, and the test set contains 42,000 independent buildings in 2416 images, as shown in Figure 7.

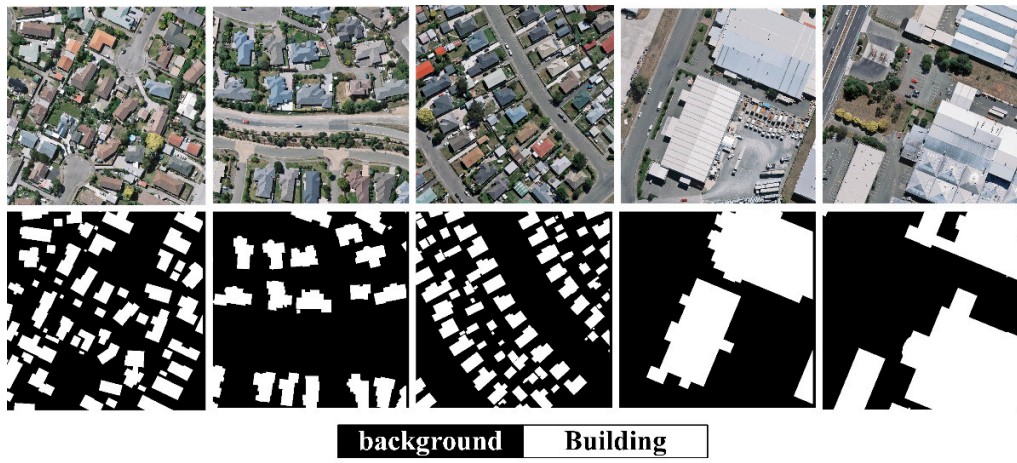

**Figure 7.** Some typical sample images of WHU building dataset and their corresponding labels.

4.1.2. Metrics

To quantitatively evaluate the accuracy of segmentation, this paper uses six metrics to evaluate the effectiveness of the network, which are Pixel Accuracy (PA), mean Pixel accuracy (mP), mean Intersection over Union (mIoU), Frequency Weighted Intersection over Union (FWIoU), mean recall (mRecall) and mean F-1 score (mF1). Among them, mIoU and FWIoU are regional evaluation metrics, and PA, mP, mReCall, and mF1 are pixel-level evaluation metrics. The calculation formulas are respectively as Formulas (1)–(6).

$$\text{PA} = \frac{\sum\limits_{c=1} (TP_c + TN_c)}{\sum\limits_{c=1}^{N} (TP_c + FP_c + TN_c + FN_c)} \tag{1}$$

$$\text{mP} = \frac{1}{N} \sum_{c=1}^{N} \frac{TP_c}{TP_c + FP_c} \tag{2}$$

$$\text{mRecall} = \frac{1}{N} \sum_{c=1}^{N} \frac{TP_c}{TP_c + FN_c} \tag{3}$$

$$\text{mF1} = \frac{1}{N} \sum_{c=1}^{N} \frac{2 \times \frac{TP_c}{TP_c + FP_c} \times \frac{TP_c}{TP_c + FN_c}}{\frac{TP_c}{TP_c + FP_c} + \frac{TP_c}{TP_c + FN_c}} \tag{4}$$

$$\text{mIoU} = \frac{1}{N} \sum_{c=1}^{N} \frac{TP_c}{TP_c + FP_c + FN_c} \tag{5}$$

$$\text{FWIoU} = \frac{1}{N} \sum_{c=1}^{N} \left( \frac{(TP_c + FN_c)}{(TP_c + FP_c + TN_c + FN_c)} \times \frac{TP_c}{(TP_c + FP_c + FN_c)} \right) \tag{6}$$

### 4.1.3. Training Details

All the experimental program codes are based on the pytorch deep learning framework. For all training samples, the mean and standard deviation values of the training set are used for normalization and the online augmentation with random rotation ($[-10°, 10°]$) and Gaussian noise ($\sigma \in [0, 1.5]$) are used for increasing the size. In the three experiments, the validation-based early stopping mechanism through monitoring the loss value, the optimizer SGD with momentum value of 0.9 and weight attenuation of 0.0001, and the Cross-entropy loss function are used by all networks [28,29]. The learning rate is initialized with 0.001 and scheduled by poly. The backbone and number of epochs settings for the three experimental datasets can be seen in Table 1. For all backbone networks, the pretrained models on the ImageNet dataset are used as the initial weight files for network training.

**Table 1.** Backbone and number of epochs settings for different experimental datasets.

|  |  | **MZData** | **Landcover.ai** | **WHU Building Dataset** |
|---|---|---|---|---|
| Backbone | FCN | | VGG16 | |
|  | Other Networks | ResNet101 | | ResNet50 |
| Number of | Total | 500 | 200 | 100 |
| Epochs | Early Stopping | 100 | 50 | 20 |

### 4.2. Ablation Studies

To verify the role of both HFE and MFF modules, based on MZData, we designed a set of ablation experiments. In the experiments, UperNet was used as the basic network. UperNet + HFE was obtained by replacing the feature extraction part in UperNet with a HFE module. UperNet + MFF was obtained by replacing the feature fusion part in UperNet with MFF module. UperNet + HFE + MFF, namely HFENet, was obtained by replacing the feature extraction part and the feature fusion part in UperNet with a HFE module and a MFF module. The backbone used for each model was resnet101, the epochs for training was 500, and the initial learning rate was 0.001.

Based on MZData, the results of evaluation metrics such as mIoU, FWIoU, PA, mP, mRecall and mF1 were obtained by experimenting with UperNet and different variants of the network were obtained after replacing its modules (Table 2).

**Table 2.** The quantitative results of HFE and MMF ablation experiments on the MZData (%).

| **Method** | **mIoU** | **FWIoU** | **PA** | **mP** | **mRecall** | **mF1** |
|---|---|---|---|---|---|---|
| UperNet | 79.78 | 87.98 | 93.42 | 88.44 | 88.19 | 88.28 |
| UperNet + HFE | 82.03 | 90.46 | 94.92 | 92.06 | 87.79 | 89.58 |
| UperNet + MFF | 80.85 | 89.27 | 94.21 | 90.41 | 88.11 | 88.81 |
| HFENet | 87.19 | 93.56 | 96.60 | 93.61 | 92.18 | 92.87 |

As can be seen from Table 2, compared with the UperNet network, whether it is the improvement of introducing HFE for feature extraction or the introduction of MFF for feature fusion, the results of various evaluation indicators have a certain improvement. In the experiments, the improvement with HFE alone is better than MFF; the effect of using two improvements (HFENet) simultaneously is more obvious. In order to verify the respective roles of the HFE module and the MFF module in the network, through the quantification of the results, the IoU evaluation index results of each category in the test set were obtained (Table 3), and the four experimental results were visualized (Figure 8).

**Table 3.** The IoU values for each category of HFE and MFF ablation experiments on the MZData (%).

| Model | Cropland | Garden Plot | Woodland | Grassland | Building | Road | Water | Bare Land |
|---|---|---|---|---|---|---|---|---|
| UperNet | 88.92 | 76.56 | 91.88 | 72.68 | 75.37 | 57.88 | 81.89 | 73.13 |
| UperNet + MFF | 89.96 | 81.35 | 94.32 | 78.83 | 69.65 | 53.34 | 83.88 | 77.63 |
| UperNet + HFE | 91.56 | 80.95 | 94.74 | 81.45 | 74.09 | 55.07 | 84.47 | 76.26 |
| HFENet | 94.66 | 86.24 | 96.18 | 85.43 | 85.66 | 65.82 | 88.35 | 82.50 |

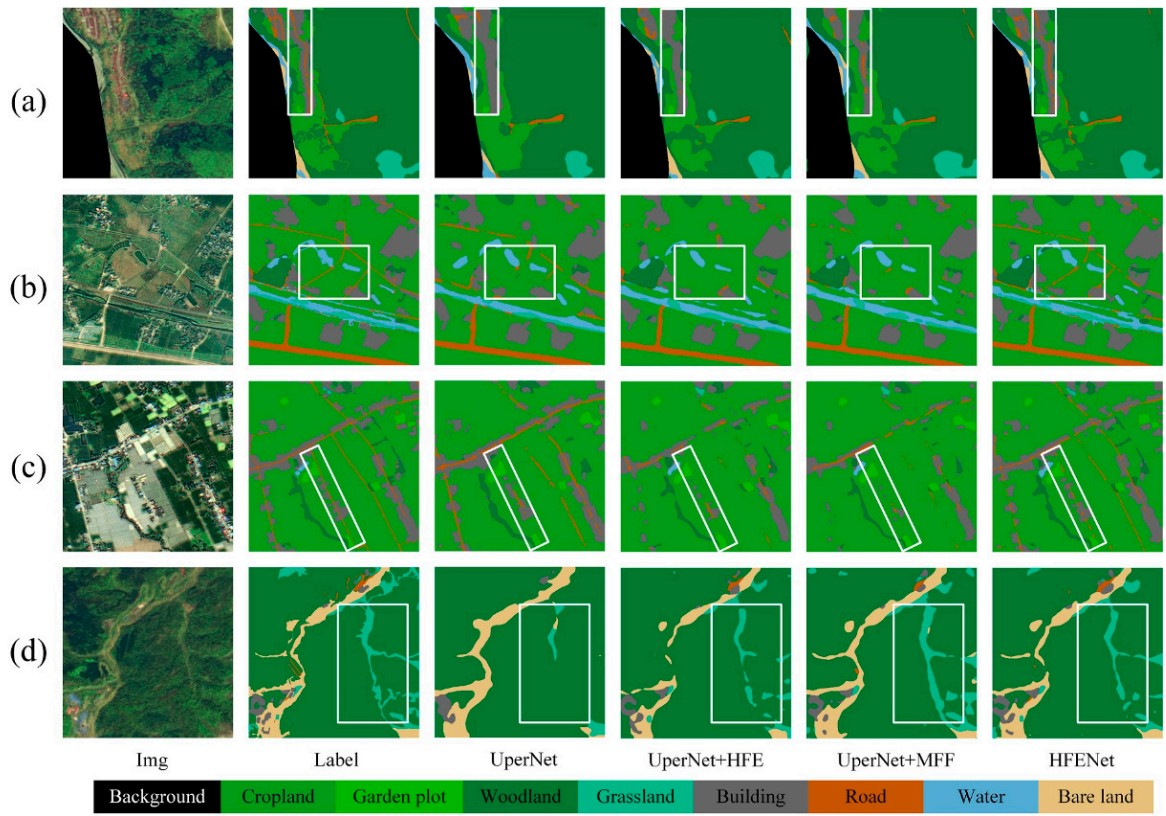

**Figure 8.** Comparison of the classification results of HFE and MMF combination on MZData. (**a**–**c**) shows the typical case that it is difficult to identify roads in small-scale features. (**d**) shows the complex scenes where woodland and grassland are easily confused. (HFENet is equivalent to UperNet + HFE + MFF).

As can be seen from Table 3, the introduction of HFE or MFF modules alone results in a significant improvement in the IoU values of confusing features such as cropland, garden plot, woodland and grassland. Because MFF assigns weights to the fused multi-scale features, and HFE module mines the relationship between channels in high-level semantic information, it can improve the expression of specific semantic information and reduce the impact of interference information on classification. However, for road and building, which have large scale differences, the IoU values not only do not improve, but also have a slight decrease. This is due to the lack of extraction of precise image location information in the case of only MFF; in the case of only HFE, the network prefers the expression of high-level semantic information in classification, resulting in incomplete representation of low-level semantic information such as spatial location. Two improvements (HFENet) are also introduced, and the IoU values are significantly improved relative to UperNet, both for confusable features and small-scale features. Especially in small-scale features (such as Road and Building), the IoU value is increased by about 10%. This is caused by the characteristics of the HFE module and the MFF module. First, the HFE module can effectively obtain the spatial location information of the images by adopting specific

extraction methods for the characteristics of different feature maps. Then, MFF can assign weights to different characteristics of the fused feature maps. Combining two modules at the same time, the network can focus not only on high-level semantic information, but also on low-level semantic information such as spatial location. From the above comparison experiments, it can be seen that the HFE and MFF modules are effective in extracting semantic information at different levels and fusing multi-level and multi-scale features.

It can be seen from Figure 8a–c that in the process of semantic segmentation, UperNet tends to ignore small-scale features, resulting in discontinuous or even unrecognized phenomena (shown in the box in Figure 8). The introduction of HFE (UperNet + HFE) or MFF (UperNet + MFF) alone not only fails to improve the network's utilization of low-level semantic information such as spatial location, but the network is more likely to ignore low-level semantic information; the introduction of both modules at the same time (HFENet) has significantly improved the recognition results. For long and narrow roads, UperNet does not recognize them at all, and HFENet can recognize them well, but there is also a phenomenon that the recognition results are discontinuous. For small buildings, the HFENet recognition results in finer contours, closer to the ideal situation.

From Figure 8d, it can be seen that UperNet cannot accurately handle the phenomena such as interlacing between features and dissimilarities in the same spectrum, which has an impact on the classification accuracy. Both UperNet + HFE and UperNet + MFF improve the network's ability to mine advanced semantic information and enhance the recognition of confusing features such as homospectral dissimilarities. HFENet not only improves the recognition ability of the network, but also is more accurate for the boundary contour information of the features.

### 4.3. Comparing with the State-of-the-Art

In order to prove the advanced nature of the method proposed in this paper, we conduct a set of comparative experiments on the landcover.ai, MZData and WHU building dataset for HFENet and the other seven most advanced landcover classification methods, i.e., U-Net [25], DeepLabv3+ [36], PSPNet [17], FCN [39], UperNet [27], DANet [18], SegNet [61], to analyze the parameters and Flops of each network and the obtained visualization and quantitative results. U-Net, DeepLabv3+, and SegNet represent encoder-decoder networks. FCN stands for fully convolutional network. PSPNet and UpperNet represent networks for pyramid pooling methods. DANet represents a network of attention mechanism methods.

#### 4.3.1. Experimental Results on MZData

The network was trained and tested on the MZData, and the results of the six quantitative evaluation metrics were calculated as shown in Table 4.

**Table 4.** The quantitative results of the state-of-the-art models on the MZData (%).

| Model | mIoU | FWIoU | PA | mP | mRecall | mF1 |
|-------|------|-------|-----|------|---------|------|
| SegNet | 77.19 | 86.22 | 92.40 | 88.15 | 85.18 | 86.55 |
| FCN | 75.63 | 85.84 | 91.99 | 85.44 | 85.09 | 85.20 |
| PSPNet | 78.47 | 87.13 | 92.83 | 87.43 | 87.34 | 87.33 |
| UperNet | 79.78 | 87.98 | 93.42 | 88.44 | 88.19 | 88.28 |
| DANet | 79.65 | 87.91 | 93.38 | 88.13 | 87.27 | 89.09 |
| DeepLabv3+ | 78.19 | 87.05 | 92.84 | 87.65 | 86.87 | 87.21 |
| HFENet (ours) | 87.19 | 93.56 | 96.60 | 93.61 | 92.18 | 92.87 |

As can be seen from Table 4, HFENet outperforms other methods in all six evaluation indicators. Compared with FCN network, mIoU is increased by 10.60 percentage points; compared with UperNet, mIoU is increased by 7.41 percentage points. To verify whether HFENet is superior to other methods in identifying small-scale features, we further counted

the IoU values of different networks for each class in the experimental results, as shown in Table 5.

**Table 5.** The IoU values for each category of the state-of-art models on the MZData (%).

| Model | Cropland | Garden Plot | Woodland | Grassland | Building | Road | Water | Bare Land |
|---|---|---|---|---|---|---|---|---|
| SegNet | 87.61 | 73.73 | 90.44 | 66.30 | 73.19 | 56.40 | 79.91 | 67.61 |
| FCN | 86.60 | 74.95 | 91.13 | 69.64 | 68.07 | 43.22 | 77.73 | 69.79 |
| PSPNet | 88.57 | 76.39 | 91.08 | 69.06 | 74.06 | 53.90 | 82.45 | 70.97 |
| UperNet | 88.92 | 76.56 | 91.88 | 72.68 | 75.37 | 57.88 | 81.89 | 73.13 |
| DANet | 88.98 | 78.72 | 91.88 | 71.45 | 75.18 | 55.47 | 83.20 | 72.30 |
| DeepLabv3+ | 88.42 | 73.81 | 91.15 | 68.76 | 74.09 | 57.07 | 81.60 | 69.11 |
| HFENet | 94.66 | 86.24 | 96.18 | 85.43 | 85.66 | 65.82 | 88.35 | 82.50 |

From the results in Table 5, it can be seen that, for the relatively small-scale ground object categories such as building and road, compared with the other six networks, the IoU value of HFENet is generally increased by about 10 percentage points. For other easily confused land object categories, such as cultivated land, grassland, and forest land, the IoU value of HFENet has also been significantly improved. In order to more intuitively illustrate the superiority of HFENet compared to other methods, we visually compare the classification results of different networks, as shown in Figure 9.

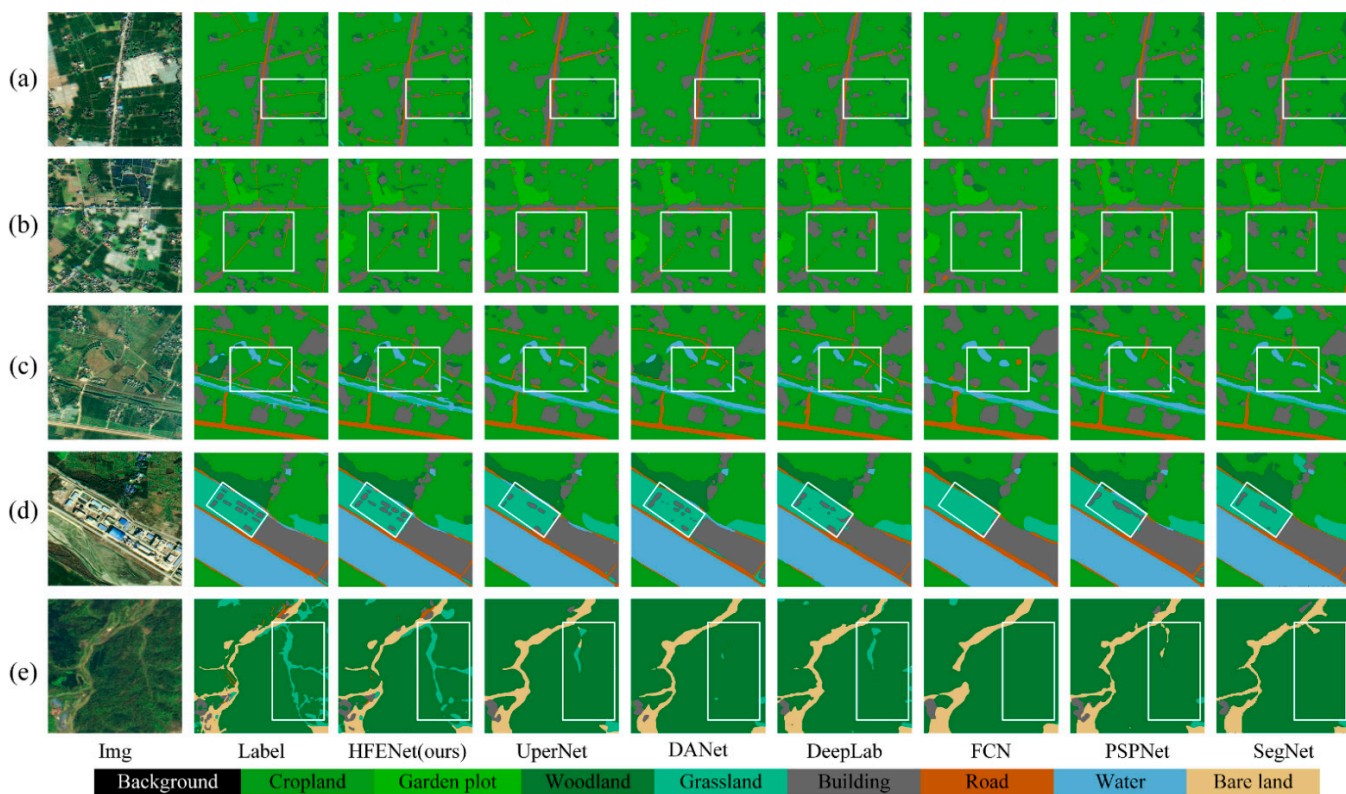

**Figure 9.** Visualization comparison of classification results of different state-of-the-art models on the MZData. (**a–d**) shows the typical case that it is difficult to identify roads and buildings in small-scale features. (**e**) shows the complex scenes where woodland and grassland are easily confused.

It can be seen from Figure 9a–c that for narrow roads, the HFENet network can identify them well, but other networks cannot identify them. In Figure 9d, for small buildings, the HFENet network identifies very well, but UperNet, DANet, and other networks do not have complete identification, and FCN does not even identify the building. In Figure 9e, for the interlaced grassland and woodland, the spectrum of the two ground objects is similar,

but the texture information is quite different, the classification effect of HFENet is obviously better than that of other networks. This shows that HFENet is able to capture the low-level semantic information contained in the underlying network using the HFE module, and then adjust the weight relationship between the high-level semantic information and the low-level semantic information through MFF, so that the network can not only separate the two categories by the low-level semantic information, but also correctly classify them by mining the high-level semantic information, thus making the goal of classification more complete.

4.3.2. Experimental Results on WHU Building Dataset

In order to further confirm the advantages of HFENet in multi-scale feature recognition, we specially chose to train and test our model and the other state-of-the-art models on the WHU building dataset [38]. The dataset is dominated by buildings, which are representative multi-scale features. According to the experimental results, we calculate the six evaluation metrics and the IoU values of the background and buildings relative to each model respectively, as shown in Table 6.

**Table 6.** Quantitative results of the state-of-the-art models on the WHU building dataset (%).

| Model | mIoU | FWIoU | PA | mP | mRecall | mF1 | IoU | |
| --- | --- | --- | --- | --- | --- | --- | --- | --- |
| | | | | | | | Background | Building |
| SegNet | 85.06 | 93.84 | 96.7 | 92.16 | 90.91 | 91.52 | 96.36 | 73.76 |
| U-Net | 87.57 | 94.92 | 97.31 | 93.78 | 92.45 | 93.10 | 97.02 | 78.11 |
| FCN | 80.55 | 91.43 | 95.08 | 85.83 | 91.98 | 88.55 | 94.54 | 66.55 |
| PSPNet | 90.95 | 96.34 | 98.09 | 95.75 | 94.52 | 95.12 | 97.88 | 84.02 |
| UperNet | 90.34 | 96.06 | 97.92 | 94.64 | 94.90 | 94.77 | 97.69 | 83.00 |
| DANet | 90.95 | 96.33 | 98.09 | 95.51 | 94.74 | 95.12 | 97.87 | 84.02 |
| DeepLabv3+ | 90.59 | 96.18 | 98.01 | 95.41 | 94.43 | 94.91 | 97.79 | 83.39 |
| HFENet (ours) | 92.12 | 96.81 | 98.34 | 95.93 | 95.67 | 95.80 | 98.15 | 86.09 |

It can be seen from Table 6 that HFENet is obviously superior to other models in terms of the classification of buildings with large scale differences, both in the six overall metrics and in the IoU of each category. The mIoU value of HFENet reached 92.12%, about 2 percentage points higher than that of DANet, PSPNet, UperNet and Deeplabv3+, about 6 percentage points higher than U-Net and SegNet; about 12 percentage points higher than FCN. In terms of building category, the IoU values of all models except HFENet are lower than 85%, and the highest is only 84.02%; the IoU value of HFENnet reached 86.09%, which is 19.45% higher than that of FCN and 2.07% higher than that of PSPNet or DANet.

To more intuitively illustrate the superiority of HFENet over other models, we visually compare the classification results of different models, and the results are shown in Figure 10.

In Figure 10a, in addition to HFENet and SegNet, other models have different degrees of misclassification, that is, some backgrounds are classified into building class; Compared with SegNet, the outline of HFENet classification is clearer. In Figure 10b,c, for large-scale buildings, the classification results of HFENet and PSPNet are more complete, while the classification results of other models are missing and relatively broken. In Figure 10d, for small-scale buildings, it can be clearly seen that other models cannot correctly classify buildings except HFENet. In Figure 10e, there is a colorful building being quite different from other buildings, which increases the difficulty of classification. As a result, only HFENet can correctly classify this building, while other models fail to do so. From Figure 10 and Table 6, it could be found that HFENet can not only correctly recognize targets in complex situations, but also improve the ability of multi-scale feature recognition.

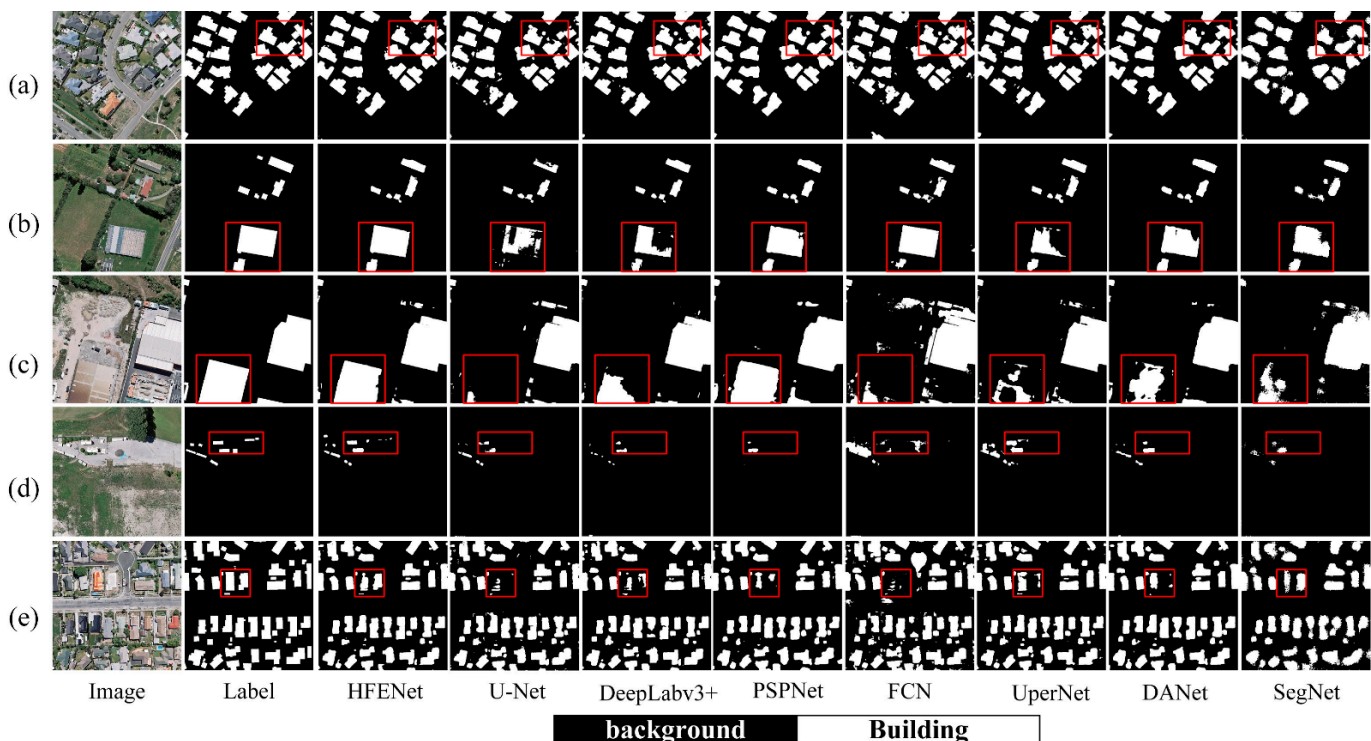

**Figure 10.** Visualization comparison of classification results of different state-of-the-art models on the WHU building dataset. (**a**–**d**) respectively shows the different cases of inaccurate classification caused by large scale difference of objects. (**e**) shows the inaccurate classification in complex scenes with large differences in features.

### 4.3.3. Experimental Results on landcover.ai

To verify the generalization of the method proposed in this paper, we train and test on the aerial image dataset (landcover.ai), and calculate six evaluation metrics for different network experimental results, as shown in Table 7.

**Table 7.** Quantitative results of the state-of-the-art models on the landcover.ai (%).

| Model | mIoU | FWIoU | PA | mP | mRecall | mF1 |
|-------|------|-------|------|------|---------|------|
| U-Net | 87.76 | 92.15 | 95.91 | 95.25 | 91.57 | 93.31 |
| Deeplabv3+ | 87.56 | 91.81 | 95.72 | 94.30 | 92.16 | 93.19 |
| PSPNet | 88.66 | 92.79 | 96.25 | 94.66 | 93.04 | 93.82 |
| FCN | 85.38 | 91.75 | 95.66 | 90.64 | 92.86 | 91.71 |
| UperNet | 88.76 | 92.56 | 96.12 | 94.00 | 93.82 | 93.91 |
| DANet | 88.34 | 92.47 | 96.07 | 93.67 | 93.67 | 93.65 |
| SegNet | 87.02 | 92.42 | 96.04 | 93.39 | 92.16 | 92.74 |
| HFENet (ours) | 89.69 | 93.21 | 96.48 | 95.21 | 93.71 | 94.44 |

From Table 7, it is obvious that HFENet outperforms other networks in all six metrics. The highest mIoU value (compared to FCN) increased by 4.31 percentage points, and the lowest (compared to UperNet) also increased by 0.93 percentage points. In order to further illustrate the advantages of HFENet in small-scale object recognition, we count the IoU values of different networks for each class, as shown in Table 8.

**Table 8.** The IoU values for each category of the state-of-art models on the landcover.ai (%).

| Model | Building | Water | Woodland | Other | mIoU |
|---|---|---|---|---|---|
| U-Net | 74.91 | 92.28 | 90.47 | 93.39 | 87.76 |
| Deeplabv3+ | 74.89 | 92.29 | 89.90 | 93.15 | 87.56 |
| PSPNet | 75.79 | 93.79 | 91.17 | 93.91 | 88.66 |
| FCN | 66.38 | 91.81 | 90.41 | 92.94 | 85.38 |
| UperNet | 77.44 | 93.01 | 90.83 | 93.77 | 88.76 |
| DANet | 76.31 | 92.48 | 90.94 | 93.62 | 88.34 |
| HFENet (ours) | 78.66 | 94.19 | 91.62 | 94.28 | 89.69 |

As can be seen from Table 8, for the building class, the IoU values of each network are below 80%, but HFENet is higher than FCN by 12.28 percentage points and exceeds UperNet by 1.22 percentage points as well. For all classes except the building class, the IoU values of each network are higher than 90% except for one (Deeplabv3+ for the woodland class), and the difference is not significant, but HFENet is higher than all other networks. In general, the IoU value of HFENet in each class is higher than other networks, and the mIoU value also has obvious advantages. The reason is that the HFENet design pays more attention to the underlying information, which improves the recognition accuracy of small-scale objects.

In order to more intuitively illustrate the superiority of HFENet compared to other methods, the results of the classification results of different networks are visualized and compared, and the results are shown in Figure 11.

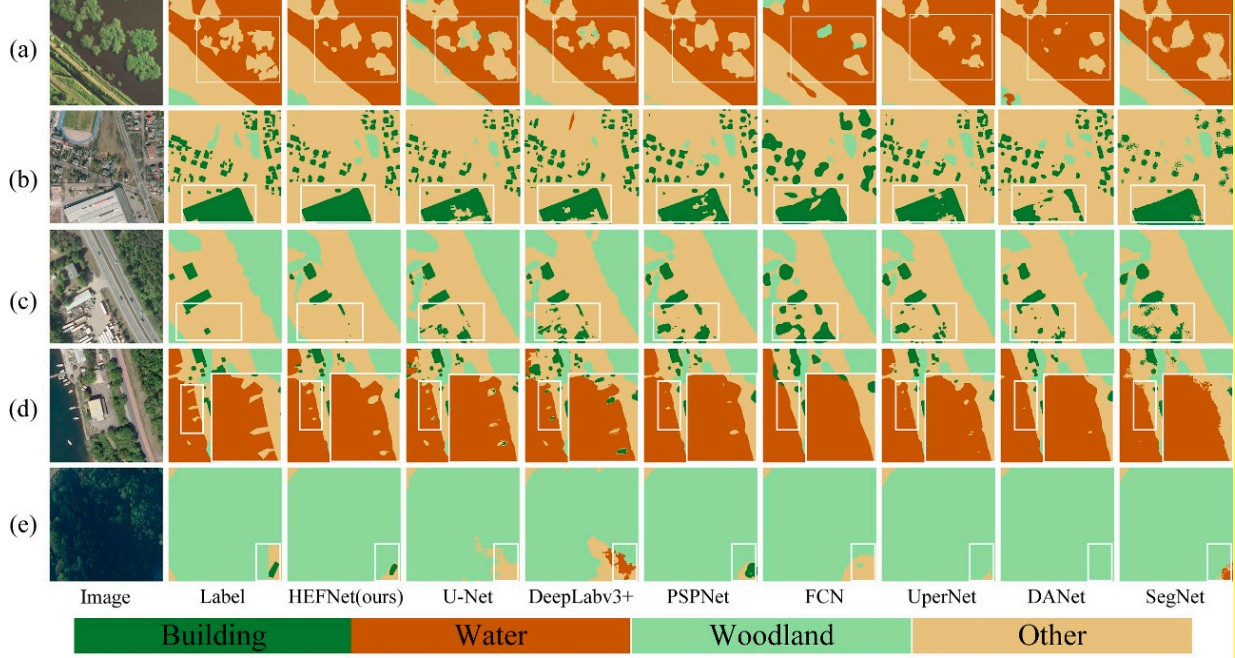

**Figure 11.** Visual comparison of classification results of the state-of-the-art models on the landcover.ai. (**a**) shows the case of inaccurate identification of object outlines, (**b**) shows the case of incomplete large-scale objects identification under multi-scale environment, (**c**) shows the case of the phenomenon of foreign objects in the same spectrum causing false identification. (**d**,**e**) shows the typical cases that small-scale objects are difficult to be identified.

In Figure 11a, for other types of objects, HFENet and PSPNet have better extraction effects than other networks; U-Net, DeepLabv3+ and SegNet not only have incomplete recognition, but also have misclassifications; DANet has obvious missed classification. In Figure 11b, for the more obvious building features, HFENet is able to identify and segment

them completely; other networks can identify buildings, but most segmentation results are incomplete, with missing, empty, or even fragmented areas. In Figure 11c, for the two objects of building and container (other category) classes, due to the small difference in color and shape, it is easy to cause confusion. HFENet can distinguish them well, but all other networks identify the container as a building. As can be seen from Figure 11d,e, for small-scale Other and Building, other networks will appear misidentified or completely unrecognized, and HFENet can correctly identify small-scale targets, although there are also minor problems of incomplete or discontinuous identification.

### 4.3.4. Comparison of Time and Space Complexity of the Models

In order to evaluate the usability of the model more comprehensively, we take the input image of $3 \times 512 \times 512$ as an example, and calculate the parameter quantity of HFENet and other state-of-the-art models as the evaluation index of space complexity, and the number of floating-point operations (flops) as the evaluation index of time complexity. The results are listed in Table 9.

**Table 9.** The Parameter and Flops of each model.

| Model | Backbone | Parameter (M) | Flops (G) |
|---|---|---|---|
| FCN | VGG16 | **190.0** | 134.27 |
| SegNet | | 53.55 | 47.62 |
| U-Net | | 30.00 | 141.31 |
| PSPNet | | 53.55 | 184.58 |
| DeepLabv3+ | ResNet50 | 59.34 | 40.35 |
| UperNet | | 107.08 | 162.78 |
| DANet | | 47.56 | **205.18** |
| HFENet (ours) | | **107.10** | 162.80 |
| SegNet | | 72.55 | 67.09 |
| U-Net | | 48.99 | 219.21 |
| PSPNet | | 70.42 | **262.48** |
| DeepLabv3+ | ResNet101 | 69.37 | 88.85 |
| UperNet | | **126.07** | 182.25 |
| DANet | | 66.55 | 283.08 |
| HFENet (ours) | | 126.09 | 182.27 |

It can be seen from Table 9 that since HFENet is improved based on UperNet, there is little difference between the two models in terms of parameter quantity and flops. From the comparison of parameter quantities alone, among all models, the FCN model with VGG16 as the backbone has the largest parameter quantity; in the network with ResNet50 and ResNet101 as backbone, the parameter amount of HFENet is the largest, but only 0.02 M higher than UperNet. From the perspective of flops, whether the model with ResNet50 or ResNet101 as the backbone, the time complexity of DANet is significantly higher than that of other models, followed by PSPNet. The time complexity of HFENet is very close to that of UperNet, and it is similar to that of FCN and U-Net, but it is significantly lower than that of DANet and PSPNet, and also significantly higher than that of SegNet and DeepLabv3++.

In view of the above, from the perspective of quantitative evaluation results and visual effects, the HFENet method proposed in this paper has achieved good results on both datasets. In the case of large differences in the scale of the same object, HFENet can accurately identify small-scale objects by obtaining low-level semantic information such as spatial location and achieve the purpose of improving the classification accuracy of small-scale objects. In the case that the ground objects are interlaced with each other, and the same-spectrum foreign objects are easily confused, HFENet can use the low-level semantic information such as texture to distinguish different ground objects, and then correctly classify different categories by mining high-level semantic information. From the perspective of algorithm complexity, HFENet has no obvious advantages over other models in terms of calculation and storage efficiency; however, compared with UperNet, in

the case of no significant change in time and space complexity, the classification effect of the model on multi-scale objects is greatly improved, which reflects the significance and value of the improved method in this paper.

## 5. Discussion

It is shown through experimental studies that the deep learning remote sensing image segmentation framework-HFENet, proposed in this paper, outperforms other state-of-the-art networks on two different datasets. In these experiments, some phenomena are worth discussing.

In order to solve the problem that UperNet does not make sufficient use of low-level semantic information and it is difficult to identify small-scale features, this paper redesigns the network by applying a hierarchical feature extraction strategy (HFE module) to the backbone network on the basis of UperNet. First, the location attention mechanism is used to focus on the underlying information to enhance the feature extraction of detailed regions and small target objects. Then, at the higher layers of the network, the interrelationships between channels are mined through the channel attention mechanism to improve the expression of specific high-level semantic information. Finally, the multi-scale information of features is obtained through the pyramid pooling module at the highest level of the network to improve the network's ability to utilize global information. However, it is found through experiments that the underlying semantic information is not well represented when only the hierarchical extraction strategy is used for feature extraction. This phenomenon is mainly due to the fact that, when fusing multi-level features, the network assigns more weight to high-level semantic information, and thus the network ignores the detailed regions contained in the underlying layer as well as small target object information. To address this phenomenon, we enhance the attention to the underlying network features by using the channel attention mechanism feature fusion method (MFF module) to reduce the risk of the underlying information being ignored. Through experiments, we found that the HFENet constructed by using both HFE and MFF modules in the entire model can achieve better performance for remote sensing image semantic segmentation.

Comparing the experimental results on three different datasets, it is not difficult to see that compared with other methods, the improvements of HFENet on MZData and WHU building dataset are significant, and the improvement on landcover.ai is relatively small. Through analysis, the most obvious improvement of HFENet's feature classification effect in MZData is on roads and buildings, and it can be seen from the visualization results that the improvement is mainly on small-scale features (such as narrow roads and fragmented buildings). This is because HFENet pays more attention to the location information extraction for the underlying network and has a high degree of attention to small-scale objects. For the WHU building dataset, HFENet can completely classify large-scale buildings and small-scale buildings. Even for some complex cases, it can classify buildings by low-level semantic information such as outline position. However, in landcover.ai, only four categories of features are segmented, and most of them belong to larger scale features, which cannot fully reflect the advantages of HFENet in small-scale target recognition. Therefore, the framework HFENet proposed in this paper can maintain the advantages of deep learning network in the recognition of ordinary scale objects in the task of remote sensing image semantic segmentation and can show better results in the refined semantic segmentation task.

## 6. Conclusions

In this paper, we propose a deep learning framework HFENet for semantic segmentation of remote sensing image landcover classification. This framework is an improvement of the UperNet framework, which mainly solves the problems that similar features in remote sensing images are easily confused and small-scale features are difficult to identify. HFENet is based on hierarchical feature extraction strategy and mainly includes two modules, HFE and MFF. The effects of HFE and MFF modules are verified by ablation

studies on the self-labeled dataset MZData. Compared with the state-of-the-art image semantic segmentation models on MZData, landcover.ai and WHU building dataset, the results show that HFENet has obvious advantages in distinguishing interlaced features with similar image features and recognizing small-scale features.

Although the HFENet proposed in this paper provides a new choice for semantic segmentation of remote sensing images, the model has no advantages in terms of time complexity and space complexity of the algorithm; In addition, the super parameters selection of deep learning methods is also a huge challenge. We spent a lot of time in the experiment to select the super parameters to ensure the performance of the model as much as possible. How to automatically adjust the parameters to achieve the best effect of the model is still worth studying.

**Author Contributions:** Conceptualization, D.W. and R.Y.; methodology, D.W.; software, D.W.; validation, Y.Q. and K.T.; formal analysis, S.L.; resources, H.L. and X.W.; data curation, H.L. and H.H.; writing—original draft preparation, D.W.; writing—review and editing, R.Y. and D.W.; supervision, R.Y.; project administration, R.Y. and J.T.; funding acquisition, J.T. All authors have read and agreed to the published version of the manuscript.

**Funding:** This research was funded by the Science and Technology Plan Project of Sichuan Province, grant number 2021YJ0369, and the Key Project of National Key R & D Program of China, grant number 2021YFB2300500.

**Data Availability Statement:** The publicly available dataset LandCover.ai can be found here: landcover.ai, accessed on 15 November 2021; the publicly available dataset WHU building dataset can be found here: http://gpcv.whu.edu.cn/data/building_dataset.html, accessed on 13 July 2022; the self-labeled dataset MZData is available on request from corresponding author.

**Acknowledgments:** The authors extend their sincere thanks to Jie Shan (School of Civil Engineering, Purdue University) for his guidance and encouragement and Po Su of Beijing Tiankai Technology Co., Ltd. for providing some RS images to support our research.

**Conflicts of Interest:** The authors declare no conflict of interest.

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
