# Peer review of "HFENet: Hierarchical Feature Extraction Network for Accurate Landcover Classification"

_remotesensing, doi:10.3390/rs14174244_

Round 1

Reviewer 1 Report

This paper proposed a novel image segmentation network, named HFENet, for mining multi-level semantic information. The workflow is presented in a logic manner and easy to read. The performance of this network was compared to other 7 state-of-the-art methods using two benchmark datasets. The results are significant with higher values in all statistical indicators. This manuscript should be considered publication in the journal, after addressing minor considerations as follows:

1)      Authors built the proposed network from UperNet. Did authors train the HFENet from scratch using two remote sensing dataset ? or did authors use the trained weights of Upernet and transfer some to HFENet ?

2)      Similar concerns to 7 state-of-the-art methods. The performance of HFENet should be compared fairly with other using similar dataset, and training strategies.

Author Response

Dear Professor:

  Many thanks for all the constructive advice and comments to our manuscript. According to your comments and suggestion, we have carefully revised our manuscript.

Yours Sincerely. Thank you very much.

Ronghao Yang

Reviewer 2 Report

This paper focus on Hierarchical Feature Extraction Network for Accurate Landcover 2 Classification. The research contents are rich and the purpose is clear. As a whole, this paper can be published as a minor revision.

(1) Some more new references should be cited.

(2) The figures should be clear enough.

(3) The methods should be logistic to simple the read.

(4) More comparisons should be provided to identify the innovation of this paper.

(5) Conclusions should be concise.

Author Response

(The authors gave the same response as above.)

Reviewer 3 Report

 It is a nice paper, but it needs a major revision as follows:

- Elaborate the main contribution of this study in the intro section, there are many similar feature extraction methods, what is the new presented in this paper. - The captions of some figures are not informative, so, improve them.

- The complexity and computation cost of this approach must be discussed.

- More comparisons must be considered. Also, parameters s setting must be discussed. Also, describe how do you guarantee fair comparison.

-See and discuss: Multi-scale geospatial object detection based on shallow-deep feature extraction; Contextual-aware Land Cover Classification with U-shaped Object Graph Neural Network (U-OGNN); PolSAR image land cover classification based on hierarchical capsule network;

- More details about your method must be added. For example, you may add pseudocodes to make it easier for the readers.

- Limitations and challenges must be furtherer discussed.

Author Response

(The authors gave the same response as above.)

Round 2

Reviewer 3 Report

The authors addressed all comments raised in the previous round. 

Thus, this version can be accepted for publication.